# Exopolysaccharide from *Porphyridium cruentum* (*purpureum*) is Not Toxic and Stimulates Immune Response against Vibriosis: The Assessment Using Zebrafish and White Shrimp *Litopenaeus vannamei*

**DOI:** 10.3390/md19030133

**Published:** 2021-02-28

**Authors:** Yenny Risjani, Nurul Mutmainnah, Praprianita Manurung, Siti Narsito Wulan

**Affiliations:** 1Faculty of Fisheries and Marine Science, Universitas Brawijaya, 65145 Malang, Indonesia; nurulmuhtar21@gmail.com (N.M.); praprianitamanurung@gmail.com (P.M.); 2Department of Agricultural Technology, Faculty of Agricultural Technology, Universitas Brawijaya, 65145 Malang, Indonesia; wulan_thpub@ub.ac.id (S.N.W.); yunianta@ub.ac.id (Y.)

**Keywords:** hemocytes, innate immune cells, phagocytic activity, respiratory burst, white shrimp, microalgae, immunomodulator, toxicity, extracellular polysaccharide, *Vibrio harveyi*, *Danio rerio*

## Abstract

Exopolysaccharides, or extracellular polysaccharides (EPS, sPS), represent a valuable metabolite compound synthesized from red microalgae. It is a non-toxic natural agent and can be applied as an immunostimulant. The toxicity test of exopolysaccharides from *Porphyridium* has been done in vivo using zebrafish (*Danio rerio*) embryonic model, or the ZET (zebrafish embryotoxicity test). The administration of extracellular polysaccharides or exopolysaccharides (EPS) from microalgae *Porphyridium cruentum* (synonym: *P. purpureum*) to shrimps *Litopenaeus vannamei* was investigated to determine the effect of this immunostimulant on their non-specific immune response and to test if this compound can be used as a protective agent for shrimps in relation to Vibrio infection. For immune response, exopolysaccharides were given to shrimps via the immersion method on day 1 and booster on day 8. Shrimp hemocytes were taken on day 1 (EPS administration), day 7 (no treatment), day 8 (EPS booster) and day 9 (Vibrio infection) and tested for their immune response on each treatment. The result shows that the EPS is not toxic, as represented by the normal embryonic development and the mortality data. In the Pacific white shrimps, an increase in the values of all immune parameters was shown, in line with the increasing EPS concentration, except for the differential hemocyte count (DHC). In detail, an increase was noted in total hemocytes (THC) value, phagocytotic activity (PA) and respiratory burst (RB) in line with the EPS concentration increase. These results and other previous studies indicate that EPS from Porphyridium is safe, enhances immune parameters in shrimp rapidly, and has the ability to act as an immunostimulant or an immunomodulator. It is a good modulator for the non-specific immune cells of Pacific white shrimps, and it can be used as a preventive agent against vibriosis.

## 1. Introduction

Vaname, the white shrimp (*Litopenaeus vannamei*), is one type of the more popular and easily cultivated shrimps. This shrimp is highly and quickly productive. Among other superiorities, it is able to live in a wide range of salinities (euryhaline), between 5 and 30 ppt, as well as being adaptable to high stocking densities, and growing well at low protein feeding levels. It entered Indonesia, according to the Decree of the Minister of Marine and Fisheries of Republic of Indonesia Number 41 of 2001, as a substitute for the tiger shrimp commodity, whose production declined due to disease attacks [1]. Nevertheless, along with the expansion of cultivation activities, diseases in white shrimp (vaname) have become increasingly common, one of which is vibriosis caused by a pathogenic bacteria *Vibrio harveyi* [2,3].

Vibriosis by *Vibrio harveyi* is still an important issue in the development of vaname shrimp cultivation. The pathogenic bacteria often attack shrimp during the nauplii, zoea, and mysis stages, and this sometimes occurs in the post-larval stage, even during maintenance in ponds at the age of 1 to 1.5 months. This disease can spread directly through the waters, as well as through direct contact between shrimps, killing the population shortly after 1 to 3 days from the initial infection [4].

Pathogen occurrences on shrimp occur faster because their immunity is different from fish. They do not have any adaptive immune system, and they only have a non-specific or innate/natural immune system, which on a cellular basis includes phagocytic activity, encapsulation, and nodule formation. They do not have certain antibodies induced by certain antigens, so they are more susceptible to outbreaks [2,5]. The emergence of disease in shrimp is often associated with poor aquatic environmental conditions, especially in Asia [6]. 

White shrimps are susceptible to outbreaks of vibriosis because they do not have any adaptive immune system, and they only have a non-specific innate immune system. In many invertebrates, the important component in an immune mechanism is the guarding process by organisms that can detect the presence of foreign objects or the emergence of foreign molecules originating from the outside of the organism. A good system must be able to stimulate defense responses from foreign molecules, including those mediated by cells. Invertebrates’ innate immunity resembles vertebrates’ immune response [7]. They do not have specific epitope immune activities and immunoglobulins, but they are able to recognize and destroy invading or parasitic microorganisms [8]. Cellular response, as a biomarker for the immune systems of aquatic animals, has been studied as an effect not only associated to bacterial infection or disease, but also related to environmental changes [9,10,11]. Invertebrates’ cell walls have proteins such as lipopolysaccharide, β-1,3-glucans (BG), and a specific protein called beta glucan binding protein (BGBP). BGBP in shrimp appears to be the main plasma protein that arises right after binding to β-glucans; β-glucan is a compound composed of polysaccharides, which react with the surface of the hemocytes and stimulate the formation of granule cells. Granule cells are one of the hemocyte cells that are responsible for the immune system in shrimps [8].

This study used exopolysaccharides (EPS, sPS) from *Porphyridium cruentum* as immunostimulants for the Pacific white shrimp. *Porphyridium cruentum* is a single-celled red microalga belonging to the class of Rhodophyceae, and it currently has another synonym: *P. purpureum* (Bory). It is capable of living freely or colonizing in marine waters, and can be cultivated in a fast life cycle. The round shaped *Porphyridium cruentum* cells are 4 to 9 µm, and almost are composed of almost 60% carbohydrates. The cells are bound in mucilage, a compound constantly excreted by the cell, forming a capsule surrounding the cell that contains polysaccharide sulfate, known as EPS in general terms or sPS in specific terms. This metabolite is one of the important components in the function of *P. cruentum* as an antioxidant, antibacterial, antiviral, and anti-hyperglycemic substance [12,13]. Various studies on the benefits of *P. cruentum* as an antibacterial have been widely carried out, but the use of exopolysaccharides from this species as an immunostimulant in *Vibrio harveyi*-infected shrimp has not been studied. Therefore, in addition to becoming the initial foundation of research related to future problems, this study is needed to determine the effectiveness of exopolysaccharides or the extracellular polysaccharide sulfate (EPS, sPS) of *P. cruentum* as a source of immunostimulants.

The term EPS in this study is used based on the previous research [14]. In fact, it is a general term of exopolysaccharides, which can also be called “extracellular polysaccharides”, based on the source of the polysaccharides’ presence as issued from the cells as a metabolite product into water media [14,15]. Based on the type of polysaccharide, the terminology used is EPS under general conditions for many diatoms species. For a specific genus like *Porphyridium*, it turns into sulphated polysaccharide, sPS [16]. Beside the genus *Porphyridium*, in general, some strains of cyanobacteria, i.e., *Cyanothece* sp., *Oscillatoria* sp., *Nostoc* sp., and *Nostoc carneum*, also secrete this compound [17].

In our previous study [15], we reported on *P. cruentum* growth and extracellular polysaccharides or exopolysaccharides production from this species. Microalgae growth was observed every day for 14 days of culturing, with the provision of 1ml/L of Fe, silicate, and vitamins, and 24-h continuous lighting. After 14 days of microalgae culturing, with densities of 15%, 20% and 25%, respectively, we had produced exopolysaccharides of 10,000 mg/L, 12,000 mg/L and 14,000 mg/L, respectively. The Fourier transform infrared spectroscopy (FTIR) test reported in our previous study showed that exopolysaccharides of *P. cruentum* consist of dominant bonds, namely phenol bonds and polysaccharide bonds [15].

Polysaccharides, one of the ingredients that can be used to produce immunostimulants, are classified as carbohydrates. They are the combination of more than six monosaccharide molecules, which can be hydrolyzed back into many monosaccharide molecules. The main sugars in this bioactive compound are xylose and galactose. The percentages of sulphate are between 7.6% and 14.6%, with protein contents of 1% to 2% and uronic acid at between 7.8% and 10% [18,19,20]. Exopolysaccharides are easily biodegradable, non-toxic, and can favor antioxidant limitations [21].

Although the exopolysaccharides have been assessed to be the non-toxic agent, their influence as an immunostimulant in immunomodulation for shrimp disease application has not yet been illustrated comprehensively, and is needed for assuredness. In this study, we use the zebrafish embryotoxicity test (ZET) method to assess the toxicity of exopolysaccharides from *P. cruentum (purpureum)*. The toxicity test using zebrafish *Danio rerio* as the model organism is increasingly used and has been widely developed. The ZET is a suitable approach to toxicity assessment for various objectives, including both toxicological risk in humans and also ecotoxicological risk in the environment. It has several characteristics that make it favorable and a good model organism, such as being easy to handle, having a high egg production rate, having sensitive responsivity and having a transparent body [22,23,24,25,26,27]. A study showed that the zebrafish genome is strikingly similar to humans [28]. It can be used for acute aquatic toxicity testing for many types of chemical compounds [29]. We use different concentrations of exopolysaccharides from *Porphyridium cruentum* as the media for the ZET, including a simple qualitative morphological view to see the teratogenic potential of this metabolite compound.

To return to the vibriosis problem in Pacific white shrimp, various preventive efforts have been undertaken, one of which is the provision of immunostimulants. The use of immunostimulant is one of the alternatives that helps to defend the organism from pathogenic infections [30,31,32]. Immunostimulant compounds can be obtained from various sources known to contain lipopolysaccharide, vitamin C, glucan, levamisole, and some microalgae. Among the many species of microalgae that are scattered in the oceans [33], one type of microalgae containing extracellular polysaccharides or exopolysaccharides for antibacterial purposes, and which is potentially a source of immunostimulants that act through antioxidant activity, is *Porphyridium cruentum* [34,35]. In this study, we suggest using *Porphyridium cruentum*, a red microalgae, to derive immunostimulants, so as to determine the effect of this bioactive compound on white Pacific shrimps’ non-specific immune response, and to test if this compound can be used safely as a protective agent against Vibrio infection.

## 2. Results

### 2.1. The Zebrafish Embryotoxicity Test (ZET)

The toxicity test of exopolysaccharides from Porphyridium was done in vivo using a zebrafish (*Danio rerio*) embryonic model, or the ZET method. Figure 1 shows the data of the number of mortalities with the concentration up to 20% of the EPS bioactive compound, and with a different exposure time, while the statistical difference is noted in Table 1. The data show that a concentration up to 15% of EPS is safe until 72 h of exposure time, or under this concentration for 96 h of exposure time, thus the concentration used in this study is appropriate. The mortality was shown immediately within 24 h of exposure time, due to shocking effect associated with initial physiological adaptation. The highest mortality was particularly noticeable with higher EPS concentrations.

As shown in the morphological development (Figure 2), in all concentration (5, 10, 15 and 20% of EPS), all zebrafish embryos developed normally, and morphologically they did not show body oddities or abnormality. Exopolysaccharides from *Phorphyridium* did not show a teratogenic effect on the embryo.

Table 2 shows the heart-beat rates of the zebrafish embryos post-fertilization, and the number of hatching eggs. The number of hatched eggs varied according to time of immersion. In general, the eggs begin to hatch after 48 h of fertilization (Hpf). Most eggs hatched between 72 and 96 h after fertilization, while the heart-beat rates of the embryos measured in 30 min showed the lowest value of 53 beats and the highest value of 86 beats.

### 2.2. Morphological View of Vibriosis in the Pacific White Shrimp (Litopenaeus vannamei)

Vaname, the Pacific white shrimp (*Litopenaeus vannamei*), infected with *Vibrio harveyi* and showing vibriosis, is shown in Figure 3. All treatments showed almost the same symptoms, whereby the shrimps experienced a change in color from the carapace and cephalotorax to the caudal parts. The whole body and the hepatopancreas organ showed smoky pale coloration, and the lateral cephalotorax and caudal fin parts were reddish orange.

### 2.3. Shrimp Immune Cells Number

The result shows that the EPS modulates all immune cells, including hemocytes. In this paper, these were expressed in total hemocyte count (THC), differential hemocyte count (DHC) (including hyaline cell numbers, and granular and semi-granular cell numbers), phagocytic activity (PA) and respiratory burst.

#### 2.3.1. Total Hemocyte Count (THC)

Total hemocyte count (THC) was observed on the first day of EPS administration, on day 7 (no treatment), on day 8, and on day 9. EPS booster administration was conducted with the immersion method using 10 ppt, 12 ppt and 14 ppt. On the ninth day, the THC values were obtained after infection with *Vibrio harveyi* 10^7^ cells/mL.

After immunostimulant addition on day 1 and day 7 (without booster), we observed changes in THC value, whereby the highest value was obtained from 14 ppt EPS, resulting in 39.9 × 10^5^ cells/mL, followed by 12 ppt EPS resulting in a THC value of 39.4 × 10^5^ cells/mL, and the lowest THC value was obtained from the control (0 ppt), resulting in 36.4 × 10^5^ cells/mL. On the eighth day, after the booster was given, or through resoaking with EPS, the highest THC value was obtained with the EPS treatment of 14 ppt, resulting in 49.2 × 10^5^ cells/mL, followed by the EPS treatment with 12 ppt resulting in 41.9 × 10^5^ cells/mL, and the 10 ppt EPS treatment resulting in 39.0 × 10^5^ cells/mL. The lowest THC value was obtained from the control (without EPS), resulting in 36.1 × 10^5^ cells/mL. After 9 days of infection, there was a decrease in THC values in all treatments. The EPS treatment of 14 ppt resulted in a decrease in THC value to 43.2 × 10^5^ cells/mL, while the treatment with 12 ppt resulted in a decrease to 39.5 × 10^5^ cells/mL, and the treatment with 10 ppt decreased the value to 32.8 × 10^5^ cells/mL. For the control, without EPS, the THC value decreased very rapidly to 23.7 × 10^5^ cells/mL (Figure 4).

#### 2.3.2. Differential Hemocyte Count (DHC)

The differential hemocyte count (DHC) consists of hyaline, which is the smallest type of hemocytes in shrimp, and two other hemocyte cells types: semi-granular and granular cells.
Hyaline cells

On day 1 the highest hyaline cell count was obtained from the 0 ppt control (37.2%), followed by 10 ppt (31.5%), 12 ppt (28.2%), and 14 ppt (24.3%). On day 7, without immunostimulant administration, there was an increase in the number of hyaline cells in all treatments, wherein the highest value was obtained from 12 ppt (37.9%), followed by 14 ppt (37.7%) and the 0 ppt control (37.3%).

On the eighth day, immunostimulant booster was given, resulting in a significant decrease in the number of hyaline cells in three treatments. The reductions of 35%, 33.3%, and 27.4% occurred for the 10, 12, and 14 ppt EPS treatments, respectively. The hyaline content of the control slightly increased to 37.5%. On the ninth day (after infection), the hyaline cell content significantly increased. The highest hyaline cell value was obtained from the 14 ppt EPS treatment (55.1%), followed 12 ppt (47.8%), and 10 ppt (43.4%). The hyaline value of the control (no treatment, 0 ppt EPS) was 41.1% (Figure 5).
Semi-granular and granular cells

The lowest semi-granular cell content on all days was obtained from the treatment with 14 ppt EPS; 11.50% was derived on day 1, 16.80% on day 7, 14.30% on day 8, and 8.10% on day 9 (Figure 6). Reductions in the cell numbers were shown with all concentration treatments. On the other hand, the treatment with the higher concentration caused a reduction in semi-granular cells.

In contrast the with-semi granular cells, granular cells increased as the exopolysaccharide concentration increased. The highest granular cell values were obtained with all the 14 ppt EPS treatments (64.20% on day 1 with immunostimulant administration, 46.60% on day 7 without immunostimulant administration, 54.60% on day 8 after the booster, and 47.20% on day 9 after infection). On the seventh day without EPS addition, the content of granular cells appeared to decrease; this indicated a reduction in the shrimp’s immunity (Figure 7).

### 2.4. Shrimp Immune Activity

#### 2.4.1. Phagocytic Activity (PA)

Figure 8 and Figure 9 show the phagocytic activity (PA) of the hemocyte cells of *L. vannamae* related to *Vibrio harveyi* infection. Phagocytosis increases in line with the increasing concentration of exopolysaccharide. On day 1 after immunostimulant administration, the highest phagocytosis level was obtained from the 14 ppt EPS treatment (22.9%), followed by the 12 ppt EPS treatment (18.7%) and the 10 ppt EPS treatment (14.6%). The lowest value was measured with the control, without EPS, which was 12.3%. On the seventh day without immunostimulant, values of 13.6% and 13.3% were obtained from the control and 10 ppt EPS treatment, respectively. The treatments of 12 and 14 ppt produced the values of 15.1% and 16.4%, respectively.

In all treatments, the phagocytosis level on day 7 was lower than that on day 1, except for the control, probably due to the lack of immunostimulant treatment. On day 8 after booster administration, the phagocytosis level was generally higher than the level on day 7, except for with the control (EPS = 0 ppt).

The use of 10 ppt of EPS concentration increased the level to 15.5%, 12 ppt of EPS concentration increased the level to 18.8%, and the 14 ppt treatment increased the level to 22.8%. A significant increase in phagocytic activity was seen on day 9 after infection, but the control and 10 ppt EPS treatments did not cause any significant increase (26.3% and 27.8%, respectively). The highest phagocytic activity was caused by the 12 and 14 ppt EPS treatments, which produced 33.8% and 36.1%, respectively.

#### 2.4.2. Respiratory Burst (RB) Activity

It was found that respiratory burst activity increases with the increment in EPS concentration. An interesting phenomenon was noticed on the ninth day post-infectious treatment with *Vibrio harveyi*, where the respiratory burst (RB) value was lower than that on the first, seventh and eighth days (0.511, 0.395, 0.537, and 0.383, respectively) (Figure 10).

## 3. Discussion

This paper describes the first study of exopolysaccharide application as a “stimulator” or “modulator” of the immune system in *Litopenaeus vannamei* in relation to *Vibrio harveyi* infection. We revealed that the EPS from *Porphyridium cruentum* (*P. purpureum*) modulates all immune cells rapidly, as shown by the total hemocyte count (THC), the differential hemocyte count (DHC), the phagocytic activity (PA) and the respiratory burst (RB), whereby increasing the EPS concentration caused a stronger effect of stimulation, and this indicated that EPS is a good modulator for the non-specific immunity of Pacific white shrimps. Regarding the toxicity of the compound for the test animals, several series of separate experiments using the ZET method also showed that the exopolysaccharides from *Porphyridium cruentum* (*purpureum*) are not toxic and can be safely administered to increase the immune system in aquatic animals.

There exist various methods, approaches and animal models to analyze toxicity via biochemical, cellular or mortality assays [10]. Indeed, the toxicity test using a rat model shows that the *Porphyridium* biomass was not toxic [36]. A separate toxicity experiment has been carried out to test if the exopolysaccharides of red microalgae are a toxic compound or not. Moreover, EPS has been evaluated to be a non-toxic organic compound, and our study using EPS did not reveal any toxicity.

This study has revealed that exopolysaccharides from *Phorphyridium* did not show a teratogenic effect on the embryo. The morphological features of the embryo did not show any abnormality. It has also shown that the heart-beat of the embryo is in a normal, healthy condition. Various studies of toxicity tests have been done using different animal models, and zebrafish has been widely used in toxicity studies [22,23,24,25,26,27]. Those studies show that the ZET method is a suitable approach for toxicity assessments with various objectives, and for assessing both toxicological risk in humans and ecotoxicological risks in the environment. It has several characteristics that make it favorable and a good model organism, such as being easy to handle, having a high egg production rate, having sensitive responsivity, and having a transparent body. A study has shown that this species has about 70% gene-similarity to humans [28], which also makes it an alternative for humans.

In our preliminary experiment of infection, we used a certain number of *Vibrio harveyi* (10^7^ cells/mL), so that the shrimps would become infected. Other previous studies reported the same bacterial cell concentration [3,32]. After infection with *Vibrio harveyi*, all individuals showed the same indicators of color change, whereby the shrimps infected experienced a change in color from the carapace to the caudal part, i.e., from a clear grey to a grey- reddish color on certain parts, and to a smoky, pale coloration on the whole body. However, no softened carapace was found until a day after infection. Softened shrimp carapace conditions occurred after three days post bacterial infection. In some cases, the signs of disease can consist of one or more indicators, i.e., lesions, reddish color change, melanization or discoloration, and loss of function of affected parts. In general, vibriosis infection shows several clinical symptoms, including, among others things, the hepatopancreas changes color to brownish red from its original black color, the body’s color changes to brownish-red that can be seen on the uropod or caudal fins, the carapace and pleopods (swimming legs) and the carapace becomes soft. If the vibriosis is very severe, the body of the infected shrimp will look interrupted at night [29,37,38].

Generally, in all treatments, the higher concentration of EPS extracted from *Phorphyridium* influenced the modulation of immune cells. This effect can be interpreted as an “immunostimulatory” effect caused by the exopolysaccharides. After infection, this effect can be assumed as “unfavorable” or, in the other words, the infection influenced the reduces number of immune cells, but the addition of EPS concentration has stimulatory effect on the shrimp’s health. The stimulation or modulation of immune cells occurs very rapidly, and this is the reason we conducted the experiments for nine days. This study is in line with other studies that had revealed that, due to bacterial infection, the hemocyte levels in invertebrates change within hours [39,40]. The cellular immune response was triggered within minutes of bacterial introduction, and the time required for the bacterial immune challenges can be seen in a hemocyte and phagocytosis assay is 24 h [41].

Decreasing and increasing total hemocyte (THC) values after bacterial infection can occur rapidly due to the body’s defense efforts. The pattern of the immune system parameters’ increases and decreases in *Vibrio alginolyticus*-infected white shrimp was recorded in the previous study, wherein the THC values began to decline from 0 to 24 h, and increased again after 36 h as a form of shrimp natural immune system recovery [39]. By increasing the number of hemocytes, the shrimp protects itself from infection, but when the infection occurs and the process of resistance to foreign matter is successful, the number of hemocytes decreases, since a large number of them die after destroying foreign macromolecules.

Hyaline cells in shrimp are important for the defense system. These cells are the smallest cell type with a ratio of high cytoplasmic nuclei and relatively few cytoplasmic granules. The number of the semi-granular cells is related to the increase and decrease in the numbers of hyaline cells, whereby semi-granular cells are formed from the advanced stages of hyaline cell development. As a result, these cells cannot develop into semi-granular cells, so this makes the number of semi-granular cells decrease.

The increase in hyaline cells is associated with phagocytic activity: when hyaline cells are infected, there will be a significant increase, as well as when an immunostimulant, which can stimulate the body’s defense activities, is given, so that hyaline cells increase as the first defense response [42]. Hyaline cells decrease after the organism is given an immunostimulant. This can be caused by granular cell formation through the process of hyaline cell maturation. Semi-granular cells are the cellular type between hyaline cells and granular cells that play an active role in the encapsulation of larger-size foreign bodies that cannot be phagocyted by hyaline [43]. The effect of EPS after infection can be assessed as unfavorable or, in other words, the infection attacks and reduces the number of semi-granular cells, and the granular cells increased at day 9, although in all treatments, the higher concentration of EPS influenced the modulation of immune cells.

Granular cells are the largest hemocyte cell type, and their nucleus is active in the process of storing and releasing prophenoloxydase and cytotoxicity systems [43]. These cells are characterized by the presence of granules in their cytoplasm. They are able to respond to polysaccharides from bacterial cell walls or β-glucan derived from fungi [44]. They play a role in phenoloxidase enzymes production for non-specific body defense activities, which are driven by the influence of immunostimulatory components, such as β-glucan, composed of polysaccharides [45]. They are involved in a fast defense response to the virus attack two hours post-infection [46].

A significant increase in phagocytic activity was noted in shrimps post infection with *Vibrio harveyi*, as seen in this study. The results above reveal that shrimps have a phagocytic activity against *Vibrio harveyi* infection, whereby the phagocytic activity itself is a reaction of the cellular defense and is an important process to maintain and eliminate microorganisms or other foreign particles that enter the body [9,39]. In general, phagocytic activity and respiratory burst both increase in line with the increasing EPS concentration.

The measurement of phagocytic activity aims to determine the level of phagocytosis that occurs through several treatments. The entry of microbial components into the body can activate the body’s defense response cellularly [9]; this can be observed through phagocytic activity, which is the main activity in the process of defending against foreign infections. Respiratory bursts were related, and they are monitored to identify the body’s defense level associated with the activity of superoxide anion (O^2−^), which was characterized by the ability of blood cells to reduce NBT (nitrobluetetrazolium). In addition, the value of RB is related to the level of phagocytosis. The higher the respiratory burst value, the better the shrimp’s defense system [43].

Respiratory burst is an advanced activity of the phagocytosis process, whereby the particles planted in phagolysosomes will be destroyed by digestive respiratory burst enzymes until free radical release occurs in phagolysosomes. In line with this study, the RB activity of turbot phagocytes increases in high concentrations of water-soluble seaweed extracts [47]. While RB activity decreased when the treatment was supplied with serum, this might cause suppressive effects on the cellular parameters. In contrast, the RB increased when the aquatic animals were under osmotic stress [48].

Various preventive efforts have been made, including, among others, using macroalgae extracts to overcome vibriosis in shrimps, and particularly in black tiger shrimps [49] and in white shrimps [50]. Algae has polysaccharide contents, which are potentially useful for various purposes [51], including, among others, EPS. EPS has been tested to have antimicrobial activities, particularly for HSV virus, types 1 and 2 (*Vaccinia* virus and *Vesicular stomatitis* virus), and two Gram-negative (*Escherichia coli* and *Salmonella enteritidis*) and one Gram-positive (*Staphylococcus aureus*) bacteria. All EPS extracts revealed a strong activity against *V. stomatitis* virus, higher than the activity of all chemical compounds tested [14]. In this study, we revealed that the EPS stimulates or modulates all immune biomarkers rapidly, and this indicates that EPS from microalgae is a good modulator for the non-specific immunity of Pacific white shrimps. In spite of the fact that the EPS can modulate the immune response of white shrimps rapidly, there is still a need for further research on the function of EPS, not only as an immunostimulant for the preventive objective, but also as a curative material with the addition of higher doses of EPS or with a longer time of treatment after Vibrio infection.

Exopolysaccharides from *Porphyridium cruentum (purpureum)* are very promising for the health of the Pacific white shrimps, as related to their immune system. Although our previous study has shown their chemical contents [15] and other studies revealed the main sugar components, which are composed of xylose and galactose, and other chemical components, such as sulphate, protein and uronic acids, in these metabolite products have been quantified [18,19,20], their structure still needs to be deeply explored. It is very important to note that for these polysaccharides, a great structure–activity dependence occurs, this being of great relevance when undertaking a structural characterization of them.

## 4. Materials and Methods

### 4.1. General

This study used an experimental method with a simple complete randomized design (CRD), using three different dosage treatments and one control, each with three replications. Red microalgae (*Porphyridium cruentum*) were obtained from Situbondo Brackish Aquaculture Center (BBAP) in East Java. The test animals were Pacific white shrimp (*Litopenaeus vannamei*) obtained from UPT Brackish Water Aquaculture Center of Bangil, East Java. The research was conducted at the Fish Reproduction Laboratory of Parasites, the Fish Diseases Laboratory, the Marine Sciences Laboratory of Fisheries and the Marine Sciences Faculty of Universitas Brawijaya, Malang, Indonesia.

### 4.2. Culture Condition

The Pacific white shrimps or vanname (*Litopenaeus vannamei*) at the young stage, aged 45 days or more, with a length of 7 cm and weight of 5 g, were maintained in aquariums. Acclimatization was done one day before treatment, using pellet feeding, and they were cultured in laboratory conditions. Temperature, pH, dissolved oxygen (DO) and salinity were controlled during the study. The daily temperature was between 25 and 26 °C, the pH was 7.5–7.8, the DO value was between 5.64 and 6.5 mg/L, and the salinity was 35 ppt.

### 4.3. Extracellular Polysaccharides Extraction

Extracellular polysaccharides (EPS) can be obtained from microalgae as a supernatant. The EPS supernatant was obtained from the centrifugation process of *Porphyridium cruentum* together with the medium of its life. Centrifugation was carried out at a speed of 10,000 rpm for 15–20 min. The acquired supernatant was separated using microalgae pellets; only the supernatant was used. The combination of all *P. cruentum* supernatants was carried out by maceration using Ethanol 96% solvent, with a solvent and media ratio of 1:0.75 *v*/*v*. The samples were stored and allowed to stand at room temperature for 72 h until white EPS deposits were formed. After 72 h, the sample was put in a water bath for 1 h at 80 °C. The samples were filtered using simple filter paper, and precipitation was carried out using cold ethanol. EPS suspensions were obtained in the freeze dryer and dialysis was carried out by resuspending the dry EPS into distilled water, and this process was carried out several times.

We have reported the chemical functional group of EPS in our previous study, which used the Fourier transformed infrared (FTIR) method to determine the chemical groups of the compounds contained in EPS as initial information about the chemical composition of EPS [15].

### 4.4. Shrimp Treatment

The test animals were white shrimps (*Litopenaeus vannamei*), 7 cm in length and 5 g in weight. The shrimps were acclimatized before treatment at a laboratory in a controlled condition. The test treatment was carried out by challenging the white shrimp (*L. vannamei*) using *Vibrio harveyi* with the density of 10^7^ cells/mL using the immersion method. The test shrimp samples were maintained by administering EPS as the immunostimulant with variations in dosages of 10 ppt, 12 ppt, and 14 ppt. Immunostimulant addition was carried out on the 1st day. The shrimps were then left alive without immunostimulant until the 7th day and were given a booster immunostimulant on the 8th day. On the 8th day (3–4 h after immunostimulant was given), the shrimps were infected with 10^7^ cells/mL of *V. harveyi* for 24 h. Hemolymph was taken from the shrimps at each treatment as the parameter test material for the study of immune cells.

### 4.5. Immune Parameters

The parameters, which included total hemocyte count (THC), differential hemocyte count (DHC), phagocytotic activity (PA) and respiratory burst (RB), were calculated using a hemocytometer with the help of a light microscope with 400× magnification. Total hemocyte count (THC), differential hemocyte count (DHC) and phagocytotic activity (PA) were based on the procedure as described in the previous settudies [9,52]. The respiratory burst activity was measured using the reduction of nitro-blue tetrazolium (NBT) assay [53].

### 4.6. Toxicity Test Using the ZET Method

To test the toxicity of EPS, another separate experimental study was performed previously. The mortality test indicates the proportion of embryonic zebrafish that die when immersed in various EPS concentrations: control, 5%, 10%, 15%, and 20% (*v*/*v*), respectively, during 24, 48, 72 and 98 h of exposure time (Appendix A). Each treatment was done with 20 embryos in triplicate. All experimental data including the morphological features and heart-beat frequency data were taken using an Olympus microscope CE-21 with 40 to 100 times magnification. Heart-beat frequency was counted using a hand tally counter and a stop watch for 30 s.

### 4.7. Data Analyses

The data were analyzed using one-way analysis of variance (ANOVA) with a confidence interval of 95%. This analysis was used to analyze differences in the average value between groups of treatments or variations obtained between test groups. Thus, if F count > F Table 5% and F Table 1%, it can be concluded that the results of this research are significantly different. Then, the test was continued with the least significant difference test (LSD) and Tukey test. Data of the toxicity test were analyzed separately by using descriptive statistics (SPSS version 16) for median and interquartile range.

### 4.8. Ethical Consideration

Approval according to the regulations on the use of animals in our study was not necessary because our research used a limited number of common shrimps frequently consumed by the wider community.

## 5. Conclusions

This study concludes that the exopolysaccharide synthesized from *Porphyridium cruentum* is a non-toxic metabolite compound. The ZET method shows that the concentration used in this study is relatively safe, supported strongly by the mortality, morphological features and heart-beat frequency data. Moreover, this biodegradable organic metabolite is very valuable as a preventive agent protecting the Pacific white shrimp from vibriosis infection. The provision of exopolysaccharides’ (EPS, sPS) immunostimulants from *Porphyridium cruentum* can rapidly stimulate the non-specific immune activity of *Litopenaeus vannamei*, with the best dose being 14 ppt. Immune activity responses were characterized by an increase in THC value, DHC (hyaline and granular cells), PA, and RB, particularly before being infected with *Vibrio harveyi*. Although post-infection, decreases were identified in all parameters, except PA, but in general all immune parameters showed increased response as the EPS concentration increased. All these results indicate that the EPS from *P. cruentum* is a good modulator for the non-specific immune cells of Pacific white shrimps, and it can be used as a preventive agent against vibriosis.

## Figures and Tables

**Figure 1 marinedrugs-19-00133-f001:**
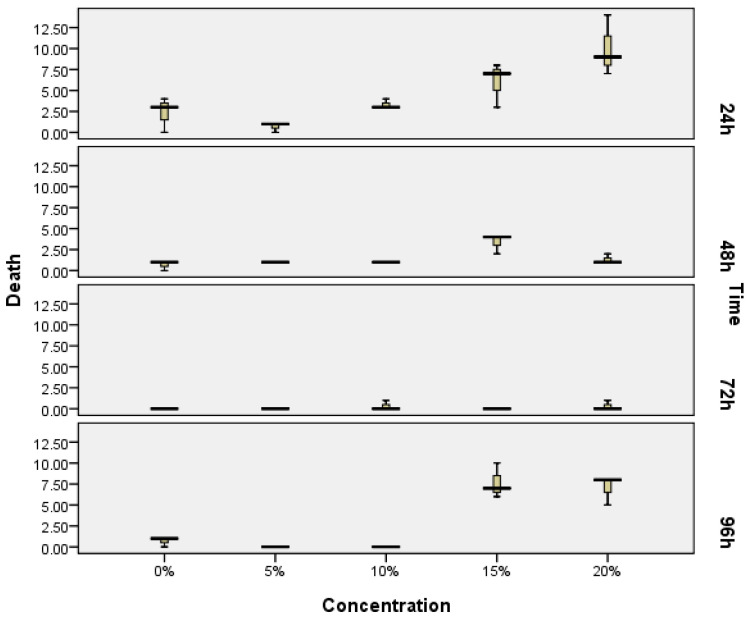
Median number of individual deaths of zebrafish embryos observed following exposure to different exopolysaccharide concentrations according to the zebrafish embryotoxicity test (ZET) method. Box represents 25th–75th percentiles; bars represent minimum and maximum values; Twenty embryos were treated in each concentration treatment (N = 20) in triplicate.

**Figure 2 marinedrugs-19-00133-f002:**
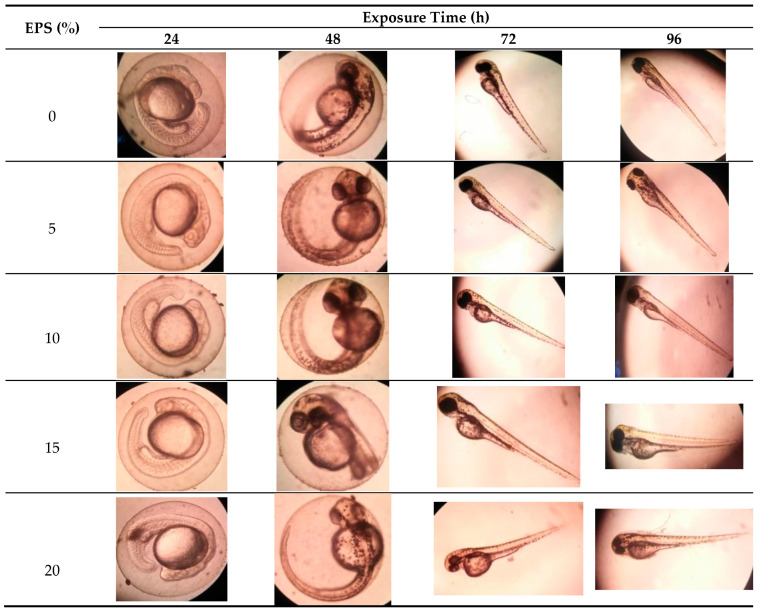
Zebrafish (*Danio rerio*) embryonic development exposed to exopolysaccharide from *Porphyridium cruentum (purpureum)* at different EPS concentrations (*v*/*v*). Photos taken under a microscope Olympus CE-21 with 100 times magnification.

**Figure 3 marinedrugs-19-00133-f003:**
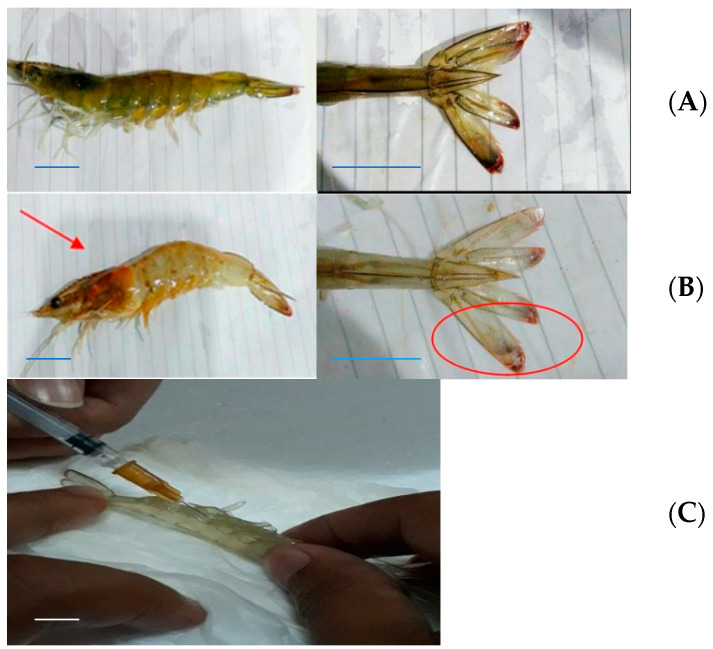
Morphology of Vaname (*Litopenaeus vannamei*). Healthy (**A**) and shrimp infected by *Vibrio harveyi* (**B**) showing vibriosis as indicated by smoky body and organ coloration, and reddish color change at the cephalotorax and caudal fin parts (arrows). Left: whole body; center: uropod. (**C**) Hemocyte preplacement with a needle. Scale bars: 1 cm. (Photos: Intan Hasanah).

**Figure 4 marinedrugs-19-00133-f004:**
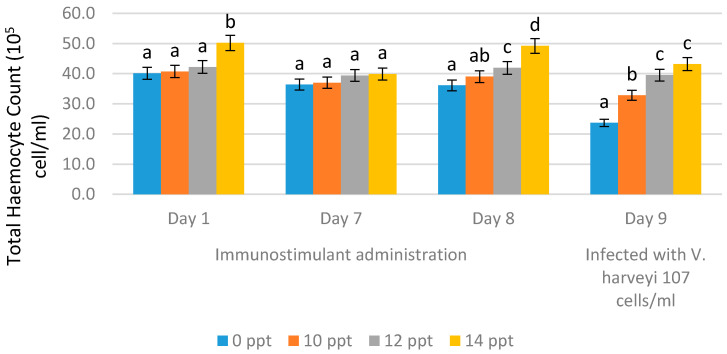
Total hemocyte count (THC) of *Litopenaeus vannamei* after EPS administration on day 1 (EPS administration), day 8 (EPS booster), day 9 (post-infection with *Vibrio harveyi*, 10^7^ cells/mL). Bars represent mean with SD, and different letters (a, b, c, d) between the bars indicate highly significant differences (*p* < 0.05).

**Figure 5 marinedrugs-19-00133-f005:**
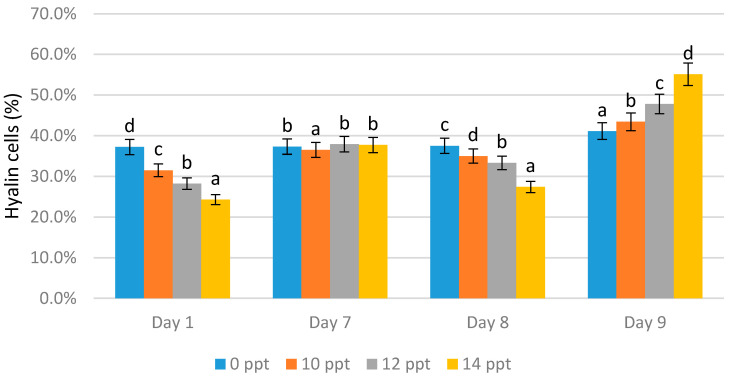
Percentage of Hyalin cells of *Litopenaeus vannamei* after EPS administration with different immersion concentrations. Day 1: EPS administration; Day 7: no treatment; Day 8: EPS booster; Day 9: post-infection with *Vibrio harveyi*, 10^7^ cells/mL. Bars represent mean with SD, different letters (a, b, c, d) between the bars indicate highly significant differences (*p* < 0.05).

**Figure 6 marinedrugs-19-00133-f006:**
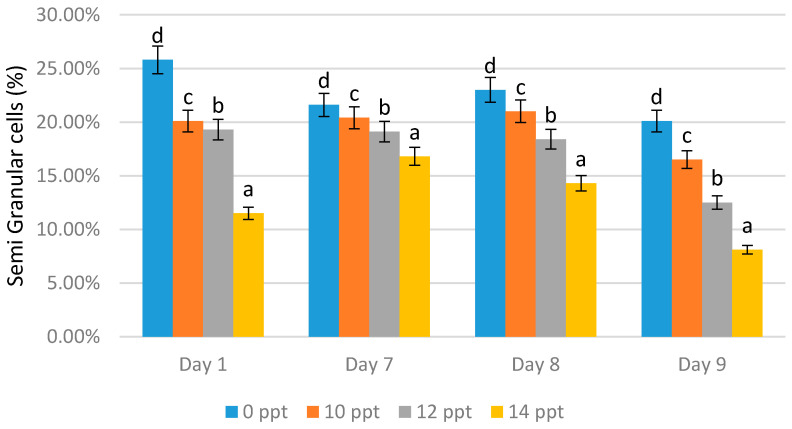
Semi-granular cells of *Litopenaeus vannamei* after EPS administration with different immersion concentrations. Day 1: EPS administration; Day 7: no treatment; Day 8: EPS booster; Day 9: post-infection with *Vibrio harveyi*, 10^7^ cells/mL. Bars represent mean with SD; different letters (a, b, c, d) between the bars indicate highly significant differences (*p* < 0.05).

**Figure 7 marinedrugs-19-00133-f007:**
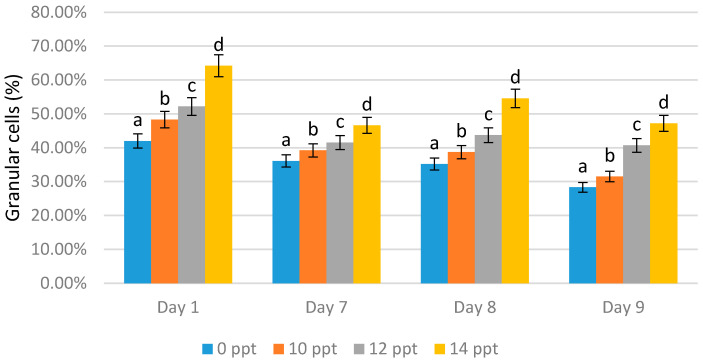
Percentage of granular cells of *Litopenaeus vannamei* after EPS administration with different immersion concentration. Day 1: EPS administration; Day 7: no treatment; Day 8: EPS booster; Day 9: post-infection with *Vibrio harveyi*, 10^7^ cells/mL. Bars represent mean with SD, different letters (a, b, c, d) between the bars indicate highly significant differences (*p* < 0.05).

**Figure 8 marinedrugs-19-00133-f008:**
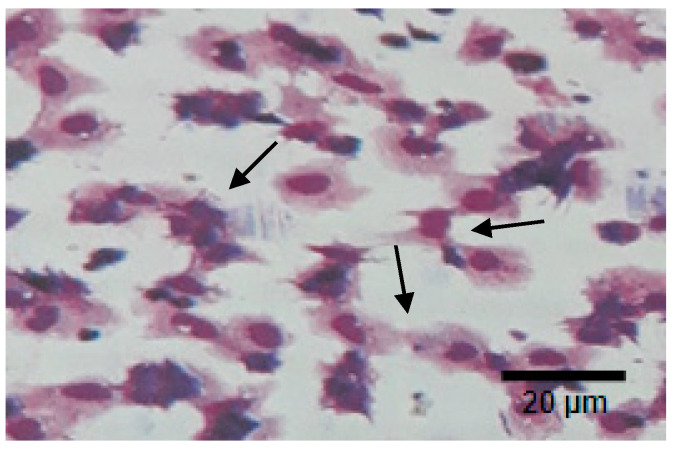
Phagocytic activity (PA) of hemocyte cells of vaname post 24 h. *Vibrio harveyi* bacterial infection. Arrows show the cells phagocyte yeast activity. Photo was taken under a light microscope (750 magnifications).

**Figure 9 marinedrugs-19-00133-f009:**
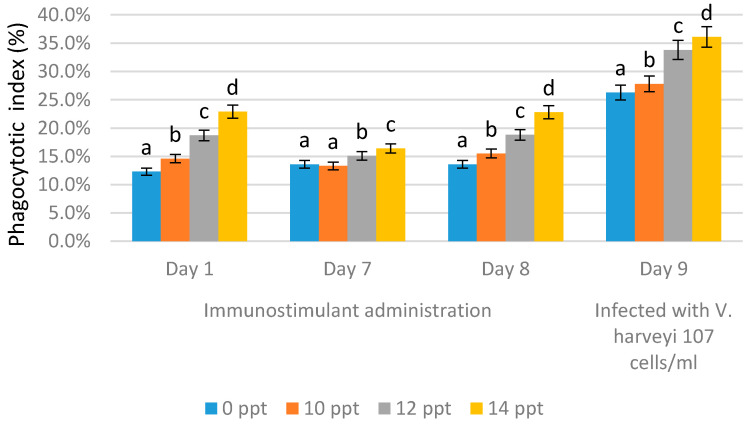
Phagocytosis activity on *Litopenaeus vannamei* after EPS administration with different immersion concentrations. Day 1: EPS administration; Day 7: no treatment; Day 8: EPS booster. Day 9: post-infection with *Vibrio harveyi*, 10^7^ cells/mL. Bars represent mean with SD, different letters (a, b, c, d) between the bars indicate highly significant differences (*p* < 0.05).

**Figure 10 marinedrugs-19-00133-f010:**
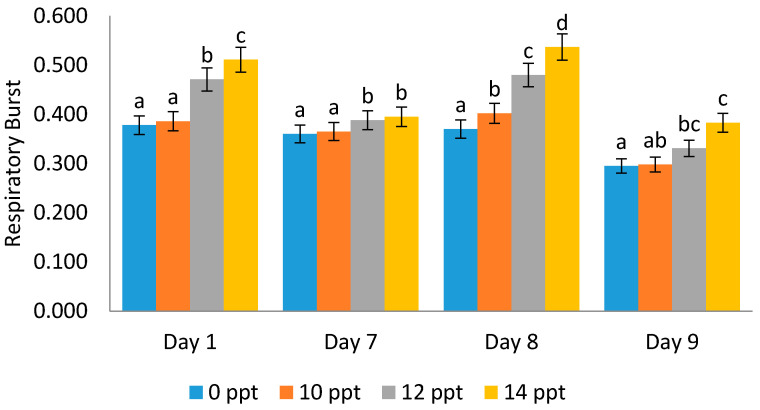
Respiratory burst activity on *Litopenaeus vannamei* after EPS administration with different immersion concentrations. Day 1: EPS administration; Day 7: no treatment; Day 8: EPS booster; Day 9: post-infection with *Vibrio harveyi*, 10^7^ cells/mL. Different letters show statistically significant difference at 0.05.

**Table 1 marinedrugs-19-00133-t001:** Least significance difference test (LSD) of the ZET at different concentrations of exopolysaccharides (EPS). Different letters notate statistically significant difference at 0.05.

EPS (%)	Mean	5%	0%	10%	15%	20%	Notation
5%	1.7						a
0%	3.7	2					a
10%	4.7	3	1				b
15%	17	15.3	13.3	12.3			c
20%	18	16.3	14.3	13.3	1		c

**Table 2 marinedrugs-19-00133-t002:** Heart-beat rates of zebrafish embryos post-fertilization in the different EPS concentrations. N represent the number of hatching eggs. Twenty eggs were treated per plate of each concentration (N = 20) in triplicate.

EPS. Concentration	Replication	Exposure Time							
		N	24 h	N	48 h	N	72 h	N	96 h
	1	0	0	0	0	12	81	19	76
0%	2	0	0	4	70	14	72	15	86
	3	0	0	0	0	7	83	15	84
	1	0	0	0	0	13	86	19	83
5%	2	0	0	4	85	17	85	18	84
	3	0	0	2	85	18	87	18	83
	1	0	0	2	72	17	70	17	84
10%	2	0	0	0	0	13	79	14	84
	3	0	0	4	78	16	76	16	85
	1	0	0	0	0	9	60	9	82
15%	2	0	0	1	80	8	79	8	87
	3	0	0	0	0	14	79	16	88
	1	0	0	0	0	8	53	10	78
20%	2	0	0	0	0	7	68	12	78
	3	0	0	0	0	0	70	5	77

## Data Availability

Data available in a publicly accesible repository.

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
