# Peer review of "Exopolysaccharide from Porphyridium cruentum (purpureum) is Not Toxic and Stimulates Immune Response against Vibriosis: The Assessment Using Zebrafish and White Shrimp Litopenaeus vannamei"

_marinedrugs, 2021, doi:10.3390/md19030133_

Round 1

Reviewer 1 Report

The document describes the effectiveness of extracellular polysaccharide sulfate from Porphyridium creuentum by stimulating the immune response of Litopenaeus vannamei against Vibrio harveyi. Although overall I am delighted with all the modifications, authors have not followed some of my suggestions. Bellow, I point out some of them again:

Now, the Introduction section is more disorderly. I reckon that the Introduction section needs an important restructuring.

From the fourth paragraph onwards, all is unclear. The aim of the manuscript is in the fourth paragraph, why? The aim should be the last paragraph from the Introduction section.

Despite abbreviating lipopolysaccharides as LPS (line 59) and extracellular polysaccharides as EPS (line 60), these abbreviations are repeated in lines 70, 88, and 105. What “FTIR” and “THC” mean? Some genus names need to be italicized. Please, modify (lines 63-67).

Results section. In figure 1, please, remove the lowercase letters within the images, they are not useful. In figures, the meaning of the a-d letters is missing in figure captions. I assume that their meaning must be related to the statistical significance, but what is their concrete meaning? Please, add the meaning in the figure captions. In line 221, replace “zoom 750” for “750x” or “750 magnifications”.

In the Materials and methods section, the second sentence from Toxicity test is a result. Please, show these results in a Figure or independent section in the Results section.

Author Response

Reviewer 1.

From the fourth paragraph onwards, all is unclear. The aim of the manuscript is in the fourth paragraph, why? The aim should be the last paragraph from the Introduction section.

Yes, the aim has been repositioned to the last paragraph and it has been revised.

Despite abbreviating lipopolysaccharides as LPS (line 59) and extracellular polysaccharides as EPS (line 60), these abbreviations are repeated in lines 70, 88, and 105. What “FTIR” and “THC” mean? Some genus names need to be italicized. Please, modify (lines 63-67).

The terms have been rewritten on the 6th-8th paragraph. We have added the meaning of FTIR and THC. We have modified the species names, they have been italicized.

Results section. In figure 1, please, remove the lowercase letters within the images, they are not useful. In figures, the meaning of the a-d letters is missing in figure captions. I assume that their meaning must be related to the statistical significance, but what is their concrete meaning? Please, add the meaning in the figure captions. In line 221, replace “zoom 750” for “750x” or “750 magnifications”.

We have removed the lowercase letters within the images of Figure 1. We have added a-d letters for the statistical signification in each graphical data. We have replaced “zoom 750” to “750 magnifications”

In the Materials and methods section, the second sentence from Toxicity test is a result. Please, show these results in a Figure or independent section in the Results section.

We try to reorganize again the manuscript because we used ZET method (Zebrafish Embryotoxicity Test), an in-vivo model for toxicity. With this type of toxicity test, we think also to change the title with the new different one.

The new title is: “Exopolysaccharide from Porphyridium cruentum (purpureum) is not toxic and stimulates shrimp immune activities against Vibrio: The Assessment using zebrafish  and  Litopenaeus vannamei

Other alternative is, using the old title without the ZET experiment included in the manuscript, but it can be presented as a supplementary data. Although, with both titles, we have proved that the metabolite is safe, as written in the discussion section (11th and 12rd paragraphs). We also discussed other finding from the rat model that the material is not toxic (Kavitha et al., 2016).

We appreciate your suggestion.

Reviewer 2 Report

The manuscript describes the effect of extracellular polysaccharides from porphyridium cruentum (purpureum) for stimulation of immune activities on white shrimp. However, it would be necessary to introduce more information about the composition of these polysaccharides. It is very important to note that for these polysaccharides, a great structure-activity dependence occurs, being of great relevance to do a structural characterization of them.  

Author Response

The manuscript describes the effect of extracellular polysaccharides from porphyridium cruentum (purpureum) for stimulation of immune activities on white shrimp. However, it would be necessary to introduce more information about the composition of these polysaccharides. It is very important to note that for these polysaccharides, a great structure-activity dependence occurs, being of great relevance to do a structural characterization of them.

The information about the composition of exopolysaccharides of P. cruentum has been described in the 6th paragraph (line 72-80). Agree with you, an important note for exopolysaccharide from P cruentum characterization is still need to be explored deeply. It has been added in the last paragraph of the discussion section.

Thank you.

Reviewer 3 Report

In my opinion, the paper is not convincing because the experimental section is poor. The manuscript is written without attention to editorial rules and scientific names, and the drafting is weak. It seems written and never reread in consideration of the too many errors present. In my opinion the paper is not acceptable for publication.

Author Response

The manuscript is written without attention to editorial rules and scientific names, and the drafting is weak. It seems written and never reread in consideration of the too many errors present.

The manuscript has been revised following editorial rules. Scientific names have been edited. 

Round 2

Reviewer 1 Report

The document describes the effectiveness of extracellular polysaccharide sulfate from Porphyridium creuentum by stimulating the immune response of Litopenaeus vannamei against Vibrio harveyi. Although overall I continue to be delighted with most the modifications, authors have not followed some of my suggestions. Bellow, I point out some of them again:

In Figure 1: please, replace “Hrs” with “h”. Furthermore, due to some bars representing the SD, I think that the values should be represented as median and interquartile range.

I must insist for the third time on the same point, please, specify what exactly means each letter in Figures 4-7. I assumed that “the letters indicate significant differences (p<0.05)” as explained in the figure caption, but what is the difference between a, b, c, and d?

Author Response

The document describes the effectiveness of extracellular polysaccharide sulfate from Porphyridium creuentum by stimulating the immune response of Litopenaeus vannamei against Vibrio harveyi. Although overall I continue to be delighted with most the modifications, authors have not followed some of my suggestions. Bellow, I point out some of them again:

In Figure 1: please, replace “Hrs” with “h”. Furthermore, due to some bars representing the SD, I think that the values should be represented as median and interquartile range.

Yes, Hrs has been replaced with “h”.  Figure 1 has been represented using whiskers-box plots with median and range. We think also to removed Table 1 because it was the same result with Figure 1. 

 please, specify what exactly means each letter in Figures 4-7. I assumed that “the letters indicate significant differences (p<0.05)” as explained in the figure caption, but what is the difference between a, b, c, and d?

The caption of those figures have been revised, "Bars represent mean with SD, different letters (a, b, c, d) between the bars indicate highly significant differences (p<0.05)."

Reviewer 2 Report

Accept in present form. 

Author Response

Thank you very much!

Reviewer 3 Report

Dear Authors,

in my opinion, although the manuscript's layout has been a little changed, it's still not very convincing (the discussion and the conclusions should be improved). 

Therefore I would suggest to reject the manuscript.

Author Response

Dear Authors,

in my opinion, although the manuscript's layout has been a little changed, it's still not very convincing (the discussion and the conclusions should be improved). 

The quality of manuscript has been increased by the supplementary data added to this paper. We changed Figure 1 graphical style and removed the unnecessary table. The layout has been revised. We have also standardized the style of the tables and of figures captions. All references list have been corrected and standardized according to editorial rules. This paper used a template from Marine drugs. The discussion and the conclusion have been improved.

Round 3

Reviewer 3 Report

Dear Authors,

I appreciated your efforts to improve the manuscript, but minor revisions are still needed. Some of them are shown in the attached file.

Author Response

Dear Reviewer,

Thank you for the revision you've sent on the pdf file. We have revised the typing errors of the manuscript based on your reviews and notes, from the titles to the references list.

In relation to Figure 2 and 8, honestly and unfortunately, we used a simple microscope which has no scale bar in it, but luckily, our student noted the magnification, 100 magnifications for the zebrafish embryos and 750 magnifications for the PA. Although there's several manual procedures how to apply scale bar manually, but in my opinion, the figure notes indicating magnifications numbers are enough to describe the figures.

The reference lists have been revised following editorial rules. The journal names used abbreviation.

Thank you.

This manuscript is a resubmission of an earlier submission. The following is a list of the peer review reports and author responses from that submission.

Round 1

Reviewer 1 Report

This article relates to the role extracellular polysaccharides from Porphyridium sp.  on shrimp Litopenaeus sp. characterizing the influence of exposure to EPS on shrimp immune response parameters. While a positive effect was observed, the current state of this article precludes any meaningful discussion of the scientific validity of this study. 

Specially, aside from a level and style of English entirely unsuitable for publication, fundamental information in the methods is missing, such as the size of the experiment and culture conditions. Furthermore, mortality rates or the influence on biomass production should also be included to show an actual benefit is observed because of this process. The authors need not only English editing, but scientific editing to improve the state of this manuscript to make it suitable for publication. 

Please see some specific comments:

EPS is listed as extracellular polysaccharide in the title, which is commonly referred to as EPS. However, in the introduction, the authors describe Extracellular Polysaccharide Sulfate, which is not the same as EPS.  The authors should consider a term such as as sulphated Polysaccharides (sPS) or polysaccharides sulfates (PSS), which are both clearly used in literature.

The mechanism through which these polysaccharides exhibit immuno-modulatory function needs to be addressed in the very brief introduction. To this end much of the discussion section belongs in the introduction, to provide the background  necessary to understand why the results are relevant .

Examples of English errors:

Level of English

Line 29 “is one type of among popular shrimps that is” should read “is one type of popular shrimp that is”

Line 30 “higher and faster” is unclear.

Line 31 “between 5 to 30 ppt, adaptable to high stocking densities, and still grow well with low protein feeding” should read “between 5 to 30 ppt, as well as being adaptable to high stocking densities, and growing well at low protein feeding levels”

Line 37 Is “Vibrious” supposed to read Vibriosis?

Line 44. “Disease attack” should read “outbreak” or “pathogen occurrence”

Line 46 “which cellularly includes phagocytic activity, encapsulation and nodule formation.”  should read "which on a cellular basis includes phagocytic activity, encapsulation and nodule formation.”

Line 59 “Mucilago” should read “mucilage”

149 2.4. Resporatory Burst (RB)....check spelling...

Reviewer 2 Report

The recent manuscript presents results on the effect of extracellular polusaccharide isolated from the red microalgae Porphyridium cruentum on some parameters of the non-specific immunity of the white shrimp (Litopenaeus vannamei). Additionally, this effect is examined after experimental infection with Vibrio harveyi which is the causative agent of vibriosis in many aquatic organisms. On the manuscript I have the following critical notes and comments: 1. It is necessary to study these parameters at a longer interval after infection, while simultaneously monitoring the presence / absence of bacterial cells, although literature indicates a decrease in the number of immune cells within a few hours after infection. These data would be not only epizootologically but also ecologically relevant. 2. The phagocytic activity as a function of immune cell maturation should also be examined in a longer period – for example day 12, 15 post LPS treatement. 3. The photo (Figure 2) is not representative enough because no significant difference is observed between "a" and "c" cells, at least when viewed with a light microscope. I would recommend submitting photos (maybe in a panel) of individual subpopulations and using an electron microscope. 4. The effect of EPS after infection can be assessed as unfavorable or, in other words, the infection attacks and reduces the number of semigranular and granular cells at day 9 (Figures 4 and 5). Can this effect be interpreted as "immunostimulatory" and how do these results explain the increased hyalin cell population. 5. How is the suspension of Vibrio harveyi standardized? 6. In the legends of all figures is necessary to be defined “a”, “b”, “c” etc.

Reviewer 3 Report

The document describes the effectiveness of extracellulaer polysaccharyde sulfate from Porphyridium creuentum by stimulating the immune response of Litopenaeus vannamei against Vibrio harveyi . Although this manuscript may be moderately innovative, it would be necessary to perform major revisions before publication:

Line 43. Please, add a little more information supported by literature about the symptoms and the physiopathology of Vibrio harveyi infection in subadults and adults of L. vannamei. Lines 235-236. Please, add the exact life cycle stages of the vaname shrimp used in this study. Lines 256-257. Please, remove this sentence. It is repeated (lines 236-237). Line 260. Why did the authors choose these concentrations of EPS and not other higher? Results section. Have the authors some photographs of the different treated groups? vaname shrimp uses to show different symptoms when it is infected by Vibrio bacteria such as black strips on both side of the lateral cephalothorax and whitish muscle or smoky body coloration. It would embellish the manuscript. Results section. In the same line of the previous comment and just as a suggestion, have the authors some histological samples of the different treated groups? It would be nice to see them. Results section. In figures, the meaning of the a-d letters is missing in figure captions. Please, add the meaning. Also, there are missing italic species names (e.g. 123, 125, 128, 130, 132, and 134). In line 120, it is necessary a change: the authors can choose between 75x or 750, but 750x is not correct. Results section. The lines 91-92 and 110-111 should be replaced in Discussion section. Results sections. Please, summarize the section 2.2. Differential Hemocyte Count. Let the graphs explain themselves and highlight the most important results. Discussion section. It is necessary to add a first paragraph which summarizes the experiment performed in this work. Lines 163-187. I consider that theses paragraphs would be better placed in Introduction section. Please, consider my suggestion. Line 190: Please, replace “at” by “from”. Lines 192-195. This sentence is not clear. Please, modify the sentence to clarify. Discussion section. The authors should consider literature related to the aim of their study and should add them: Fish Shellfish Immunol. 2011 Jan;30(1):389-96 and Fish Shellfish Immunol. 2016 Apr;51:346-350. Some genus name need to be italicized. Please, modifiy (lines 285, 287, and 289).